# MISFP-Growth: Hadoop-Based Frequent Pattern Mining with Multiple Item Support

**Chen-Shu Wang [1,*] and Jui-Yen Chang [2]**

[1]   Department of Information and Finance Management, National Taipei University of Technology,
    Taipei 10608, Taiwan
[2]   Department of Management Information System, National Chengchi University, Taipei 11605, Taiwan;
    ketrelo0225@gmail.com
*   Correspondence: wangcs@ntut.edu.tw; Tel.: +886-2-2771-2171 (ext. 2355)

**Abstract:** In practice, single item support cannot comprehensively address the complexity of items in large datasets. In this study, we propose a big data analytics framework (named Multiple Item Support Frequent Patterns, MISFP-growth algorithm) that uses Hadoop-based parallel computing to achieve high-efficiency mining of itemsets with multiple item supports (MIS). The proposed architecture consists of two phases. First, in the counting support phase, a Hadoop MapReduce architecture is employed to determine the support for each item. Next, in the analytics phase, sub-transaction blocks are generated according to MIS and the MISFP-growth algorithm identifies the frequency of patterns. To facilitate decision makers in setting MIS, we also propose the concept of classification of item (COI), which classifies items of higher homogeneity into the same class, by which the items inherit class support as their item support. Three experiments were implemented to validate the proposed Hadoop-based MISFP-growth algorithm. The experimental results show approximately 38% reduction in the execution time on parallel architectures. The proposed MISFP-growth algorithm can be implemented on the distributed computing framework. Furthermore, according to the experimental results, the enhanced performance of the proposed algorithm indicates that it could have big data analytics applications.

**Keywords:** big data analytics; Hadoop MapReduce parallel computing; frequent pattern discovery; multiple item support

## 1. Introduction

Big data analytics applications have drastically changed our daily life. The gradually maturing sensor technology and ubiquitous use of various mobile devices generate increasing amounts of data, which is expected to continue increasing by 35% annually [1]. Interest in data-driven decision making is growing in response to escalating data generation, and a large variety of big data analytics applications have been emerging accordingly. Developing models to find critical information and analyzing value from big data have become the subject of deep exploration and intense discussion [2–4].

Big data analytics and applications are interesting and important prominent topics [5]. Identifying frequent itemsets in large transactional databases, known as frequent patterns (FP), and the FP of association rules can be valuable to decision makers for setting up strategies [6,7]. Among those commonly-applied FP-mining algorithms, the Apriori algorithm is regarded as a classic method [8,9]. The Apriori algorithm uses a bottom–up approach to generate candidate itemsets. As the quantity of data is extremely large, the bottom–up approach becoming lower processing efficiency. To overcome the limitations of the traditional Apriori algorithm, Han et al. proposed an FP-growth algorithm that included two phases: constructing an FP-tree and mining the FP-tree that would be more

efficient [10,11]. For both Apriori and FP-growth algorithms, it is important to set reasonable minimum support values [12,13].

However, obviously all items in the database vary widely from many perspectives such as their price or profit. Generally, if only one minimum support is used for FP mining, those high-profit items mostly with lower selling frequencies, such as refrigerators and smartphones, would not be considered as frequent patterns [14]. Furthermore, lower minimum support would lead to the generation of many meaningless association rules, which would complicate the decision-making process [15]. Therefore, using a single minimum support for all items in the database is insufficient for FP analysis. Single item support cannot comprehensively address the complexity of items in large datasets. Thus, it is important and necessary to consider multiple item supports while big data analysis is applied in practice. Recently, several studies have proposed the concept of multiple item supports (MIS) to address the trade-offs required by decision makers [16,17]. For FP mining research, applying MIS to FP mining warrants further analysis. In addition, a large amount of research has attempted to improve the algorithmic efficiency of FP mining when working with big data. Apache's Hadoop system applies MapReduce programming to solve these problems encountered in processing big data [18]. Using Hadoop MapReduce to implement the parallelization of traditional association rule mining approaches, such as the Apriori and FP-growth algorithms, was propose to improve the overall performance of frequency pattern mining [19]. For illustration, the k-phase parallel Apriori algorithm was proposed to identify k-frequency items in k-scans using Hadoop MapReduce [20].

To optimize FP mining, this study proposes a multiple item support frequent pattern (MISFP)-growth algorithm, which mines association rules from FP by using multiple item support. Furthermore, to improve the efficiency of the analysis, the proposed algorithm is deployed on Hadoop MapReduce architecture. Section 2 summarizes and discusses the related researches. Section 3 details the proposed algorithm of Hadoop-based MISFP-growth algorithm, and Section 4 demonstrates the proposed MISFP-growth algorithm with an example implementation. Finally, three experiments are implemented to validate the accuracy and performance of the proposed Hadoop-based MISFP-growth algorithm. According to the experiment results, the proposed algorithm achieved enhanced performance that is viable for practical applications in big data analytics.

## 2. Literature Review

This section summarizes association rule mining algorithms with single support and multi-support, and the Hadoop MapReduce architecture is explained.

Association analysis attempts to determine the frequency patterns consisting of items in a dataset. The most well-known association mining algorithm is the Apriori algorithm [8,9], which is developed to scan transactional databases iteratively and generate candidate frequent itemsets. Then, a threshold known as the support filters items to form the candidate itemsets based on their frequency of occurrence. When no additional candidate itemsets can be generated, the frequency patterns (FP) and associated rules are constructed for decision makers' reference. To improve the efficiency of FP mining, many researchers have attempted to reduce the effort for repeated database (DB) scanning. Thus, the FP-growth algorithm [10] emerged to facilitate FP mining by scanning the DB twice and without generating candidate itemsets. Both the Apriori and FP-growth algorithms have been shown to successfully mine frequent patterns.

However, single item support cannot comprehensively address the complexity of items in large datasets. To apply the same single support to all items is unreasonable and insufficient. For this reason, since 1999, a considerable amount of research has focused on the MS–Apriori algorithm. The MS–Apriori algorithm applies the concept of multiple item support values to the traditional Apriori algorithm. Setting different values of support, the multiple item support (MIS) can realize decision makers' demands for highly detailed analysis [17]. The first step in MISFP mining is to pre-formulate the MIS for each item and sort all items in ascending order of magnitude. When using the Apriori algorithm in FP mining, the dataset is scanned to obtain candidate itemsets.

Mining association rules with multiple item support (MIS) has become an imperative research topic. Hu and Chen (2006) proposed the Complete Frequent Pattern (CFP)-growth algorithm to improve the FP-growth calculation mechanism [21]. The CFP-growth algorithm creates a MIS-tree that stores key messages with frequent item patterns. To improve the efficiency of the CFP-growth algorithm, the MISFP-growth algorithm discards two stages of the FP calculation, post-pruning and reconstruction [22]. The MISFP-growth algorithm finds the minimum value of the multiple threshold values (MIN-MIS) and then drops the items that with less item support than the MIN-MIS. Thus, the search space is reduced. A comparison of FP mining algorithms is given in Table 1.

**Table 1.** Comparison of frequent patterns mining algorithm.

| Algorithm | Criteria | Algorithm Description |
|-----------|----------|----------------------|
| Apriori [8] | Single support | Generates candidate itemsets by scanning the database repeatedly to successfully mine the frequent patterns. |
| FP-growth [10] | Single support | Scans the database only twice and without generating the candidate itemsets. Creates an FP-tree from which to mine the frequent patterns. However, when the MIN updates, the FP-tree must be reconstructed. |
| MSApriori [17] | Multi-support | Sets different thresholds of support to identify more rare items. However, candidate itemsets are repeatedly created and this increases memory requirements and reduces performance. |
| CFP-growth [21] | Multi-support | The arrangement of the position and order of items in the MIS-tree can be repeatedly adjusted by tuning the MIS. This method can increase search space requirements. |
| MISFP-growth [22] | Multi-support | Improves efficiency by discarding two major steps, post-pruning and reconstruction. |

Apache Hadoop is a cluster system of open source software framework composed of a series of function modules. The cloud service platform can store and manage big data, and is characterized by scalability, reliability, flexibility, and cost-effectiveness. The platform implements MapReduce distributed parallel computing architecture on the Hadoop Distributed File System (HDFS), which efficiently analyzes and stores large datasets [23,24]. The primary Hadoop operation applies the concept of "divide and conquer." Through the map and reduce functions, the problem data from each node is decomposed into several smaller blocks for calculation. Then the calculation results from all nodes are transferred, collected, and arranged. One application of MapReduce incorporates the Apriori algorithm to understand the purchase requirements of customers and attract clients from competitor e-commerce websites [25].

In summary, this study proposes advantages of using the Hadoop's parallel computing framework to overcome the drawbacks of existing single support association rule mining methods, and expresses the characteristics of each item. Using the MISFP-growth calculation method, we realize a two-stage solution that efficiently mines frequent patterns and association rules.

## 3. Hadoop-Based Frequent Pattern Mining Architecture

To enhance the efficiency of FP mining with multiple item support, we propose an architecture (named the MISFP-growth algorithm) based on a distributed computing environment, known as Hadoop MapReduce. The main idea of the MISFP-growth algorithm is to split the transaction datasets into sub-transaction DBs according to item support, and then to analyze them respectively. More specifically, transactions that contained items with identical item support would be grouped to form a sub-transaction DB. Then, the FP-growth algorithm constructed an FP-tree from these sub-transaction DB and then mine the frequency patterns. To further improve efficiency, we implement the MISFP-growth algorithm on distributed architecture, which enables individual analysis of the sub-transaction DBs. Finally, the analysis results from each reducer were further aggregated to generate association rules for the entire dataset. The proposed Hadoop MapReduce-based MISFP-growth

algorithm consists of two phases, a counting support phase and an analytics phase, executed in seven steps, as depicted in Figure 1, and detailed below.

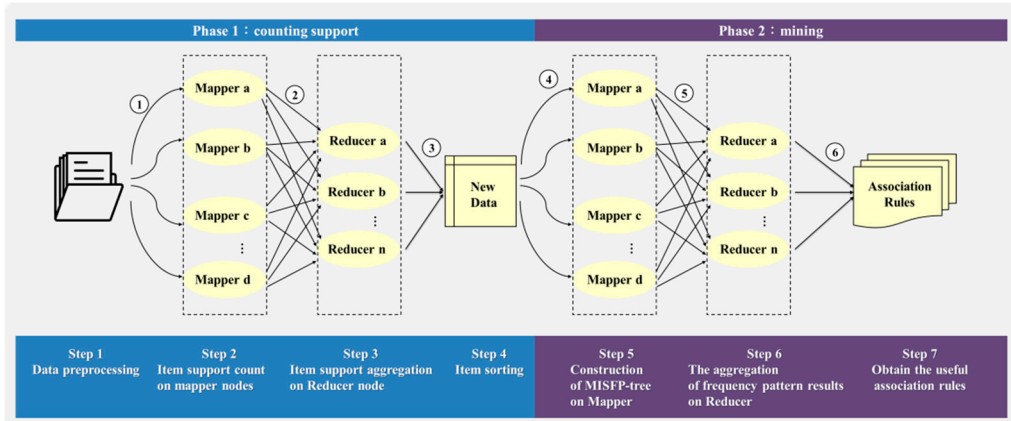

**Figure 1.** Hadoop-based frequent pattern growth mining architecture.

Step 1: data preprocessing. Step 1 cleans the original transactional DB and enables the decision maker to set multiple item supports. As shown in ① of Figure 1, the concept of "divide and conquer" is applied to the original transactional DB, which is divided into several blocks. Then, each data block is assigned to different mapper nodes for further item support counting.

Step 2: item support counting by mapper nodes. Step 2 calculates the actual frequency for each item. As ② of Figure 1 illustrates, each mapper node scans the sub-transaction DB for item frequency statistics. In the traditional FP-growth algorithm, this operation is exceedingly time consuming. As expected, the proposed Hadoop-based architecture can release such loading efficiency because the item support counting occurs in parallel. Then, the counting results from the mapper nodes are shuffled to the reducer nodes in Step 3 for further aggregation.

Step 3: item support aggregation by reducer nodes. Step 3 scans the support of all items from the mappers to find the minimum of the multiple item supports (MIN-MIS). As shown in ③ of Figure 1, each item support is compared with the MIN-MIS, if the item support is less than MIN-MIS, then the item is discarded.

Step 4: item sorting. The transaction is sorted in ascending order according to the support value and formed into new data groups. As shown in ④ of Figure 1, Step 4 generates new sub-transaction DBs based on item supports, assigning transactions from itemsets with identical support to the same sub-transaction DB and corresponding mapper nodes.

Step 5: construction of MISFP-trees by mappers. Step 5 builds a conditional MISFP-tree for each suffix item using the MISFP-growth algorithm. For each sub-transaction DB, the mappers in ⑤ output the frequency patterns that are greater than MIN-MIS, and send them to the reducer nodes for further aggregation.

Step 6: the aggregation of frequency pattern results by reducers. In Step 6, the reducer nodes check the merged results for duplicates and remove the repeated items from the output results.

Step 7: obtain the useful association rules. As shown in ⑥ of Figure 1, the MISFP-growth algorithm then discovers association rules by the mining FP-tree.

Item support exerts a substantial influence on the results of FP mining. It is difficult but crucial to set appropriate item support thresholds. Often, item support is either determined subjectively by decision makers or it is established by trial and error through recurring adjustments. However, as the dataset is large, such a readjustment process becomes time-consuming with lots of loading.

To resolve the constraints and complications of multiple item support setting, we propose the concept of classification of items (COI), which categorizes items of higher homogeneity into the same class. The main idea of COI is that the support of a specific item is equal to the support of a particular

class that the item belongs to. The concept of COI enables decision makers to set support for different items by setting class support individually. The algorithm and parameter definition of COI are listed in Figure 2 and Table 2. The support of *j*th item is obtained by: $\sum X_{ij} \in \{1, n+1\}, \forall_j = 1, 2, \dots, m$.

**Table 2.** Parameter definition of "classification of items (COI)".

| Parameter | Description |
|---|---|
| $C_i$ | $C_i$ represents the *i*th product class, where $i = 1, 2, \dots, n$. |
| $SC_i$ | $SC_i$ represents the support for the *i*th product class, $i = 1, 2, \dots, n$. |
| $I_j$ | $I_j$ represents the *j*th item, $j = 1, 2, \dots, m$. |
| $SI_j$ | $SI_j$ represents the support for the *j*th item, $j = 1, 2, \dots, m$. |
| $X_{ij}$ | $X_{ij}$ is a binary variable that maps product class and item. |

```
                                          10   FOR ( j = 1 to m) {
   Iⱼ                                      20       FOR ( i = 1 to n) {
Cᵢ    I₁    I₂   ...    Iₘ                 30           IF ( Xᵢⱼ == 1) {
C₁   X₁₁   X₁₂   ...   X₁ₘ                 40               SIⱼ = SCᵢ ;
C₂   X₂₁   X₂₂   ...   X₂ₘ                 50           }
...   ...   ...   ...   ...                60           ELSE {
Cₙ   Xₙ₁   Xₙ₂   ...   Xₙₘ                 70               SIⱼ = assigned manual SCᵢ ;
                                          80           }
                                          90       END i ;
                                          100      }
                                          110  END j ;
                                          120  }
```

**Figure 2.** Pseudo code of "classification of items (COI)".

The pseudo code of COI is shown in Figure 2. For illustration, assume that ten items (including: glass cleaner, shower gel, chocolate, whiskey, cleaning rags, cleansing milk, chewing gum, broom, shampoo, and popcorn) are represented by $I_1, I_2, \dots, I_{10}$, as shown in Table 3. There are three support levels for the product classes: $C_1$: home cleaning = 0.1, $C_2$: beauty cosmetic = 0.15, and $C_3$: snacks = 0.2. Take $I_1$ as an example: as $X_{11} = 1$ represents $I_1$ is categorized as the $C_1$ class, then the item support of $I_1$ is equal to 0.1. Conversely, looking at $I_4$ as an example: as $X_{14} = 4$, from $n + 1$, this means that $I_4$ does not belong to any product class. Therefore, decision makers must manually define item support, in this example, the item support value of $I_4 = 0.05$. Thus, COI can quickly set the multiple item support thresholds. In this example, COI categorized items as follows: $\{I_1, I_5, I_8: C_1 = 0.1\}$, $\{I_2, I_6, I_9: C_2 = 0.15\}$, $\{I_3, I_7, I_{10}: C_3 = 0.2\}$, and $\{I_4: C_4 = 0.05\}$.

**Table 3.** Demonstration of COI.

| $I_j$ / $C_i$ | $I_1$ | $I_2$ | $I_3$ | $I_4$ | $I_5$ | $I_6$ | $I_7$ | $I_8$ | $I_9$ | $I_{10}$ |
|---|---|---|---|---|---|---|---|---|---|---|
| $C_1$ | 1 | 0 | 0 | n+1 | 1 | 0 | 0 | 1 | 0 | 0 |
| $C_2$ | 0 | 1 | 0 | 0 | 0 | 1 | 0 | 0 | 1 | 0 |
| $C_3$ | 0 | 0 | 1 | 0 | 0 | 0 | 1 | 0 | 0 | 1 |
| $\sum X_{ij}$ | 1 | 1 | 1 | 4 | 1 | 1 | 1 | 1 | 1 | 1 |
| Result | $C_1$ | $C_2$ | $C_3$ | *1 | $C_1$ | $C_2$ | $C_3$ | $C_1$ | $C_2$ | $C_3$ |

[1] Note: * assigned manual.

Finally, confidence is used as an indicator to validate the meaning and reference values of the mining results [26,27]. However, higher confidence does not necessarily indicate that the rule represents a strong correlation. To this end, we used the *lift* indicator to evaluate the correlation and accuracy between items and association rules [28].

The explanation of *lift* is as follows: the statement *lift(X,Y) > 1* means item *X* has a positive correlation with item *Y*; if *lift(X,Y) = 1*, this means that item *X* is not related to item *Y*; if *lift(X,Y) <* 1, then item *X* is negatively correlated with item *Y*. Let *P(X)* be the probability that item *X* in the database. Let *P(X∪Y)* be the probability that the database contains both item *X* and item *Y*. Let *P(X∪Y) = P(X)×P(Y)* indicate that item *X* and item *Y* are independent of each other. If the previous statement is not true, then item *X* is related to item *Y*. The *lift* correlation indicator is detailed in Equation (1):

$$lift(X,Y) = \frac{P(X \cup Y)}{P(X)P(Y)} = \frac{Sup.(X \cup Y)}{Sup.(X)Sup.(Y)} = \frac{Confidence(X \rightarrow Y)}{Sup.(Y)} \tag{1}$$

## 4. Hadoop MapReduce-Based MISFP Growth Algorithm

This section demonstrates the proposed MISFP-growth algorithm with an example implementation. Continuing the example given in the previous section, the transactional database as shown in Table 4a,b lists the actual support values and MIS of each item.

**Table 4.** Transaction database, actual support and multiple item supports (MIS) of each item.

| (a) | | (b) | | | (c) | |
|---|---|---|---|---|---|---|
| Transaction ID. | Transaction | Items. | Support | MIS | TID. | Trans. (ordered) |
| 1 | B, A, C | A | 6 | 4 | 1 | A, B, C |
| 2 | E, D, H, G, I | B | 6 | 4 | 2 | E, G, H, I |
| 3 | C, A, B, J | C | 5 | 4 | 3 | A, B, C, J |
| 4 | E, C, G | D | 1 | 5 | 4 | C, E, G |
| 5 | B, A, J, I | E | 4 | 3 | 5 | A, B, I, J |
| 6 | A, B | F | 1 | 4 | 6 | A, B |
| 7 | G, C, F, E, J | G | 4 | 3 | 7 | C, E, G, J |
| 8 | B, A, G | H | 2 | 2 | 8 | A, B, G |
| 9 | H, A, I | I | 3 | 2 | 9 | A, H, I |
| 10 | C, E, B | J | 3 | 2 | 10 | B, C, E |

As illustrated in Figure 3, first, the MISFP-growth algorithm scans the transactional database once to find the support value and determine the MIN-MIS = *MIN{MIS(A), MIS(B), . . . , MIS(J)}* = 2 as shown in Table 4b. Next, the MISFP-growth algorithm sorts item {A} to item {Z} in ascending order according to item support to satisfy MIN-MIS. Therefore, items {D} and {F} are removed because their support is less than MIN_MIS. The results are presented in Table 4c. The MISFP-growth algorithm splits the transaction dataset into subgroups according to MIS values and assigns the subgroups to different mapper nodes. Each mapper node then scans the assigned sub-transaction DB to obtain the support value of each item and construct an MISFP-tree. This process resembles that of the frequency pattern tree constructed by the FP-growth algorithm.

For example, a given mapper node handles the sub-transaction DB named Block "A" for which item support is two. The first root is created and labeled with a null node of the MISFP-tree. Then, additional nodes are inserted for the leading itemsets {E, G, H, I} of Block A. Figure 4 displays all the node links of each item.

Each mapper node constructs an MISFP-tree and conducts frequency pattern mining. Using the node links of item {J} as an example, we establish a conditional pattern subtree with the following conditional pattern base: {A, B, I :1}, {A, B, C :1}, and {C, E, G :1}. Nonetheless, the MIS of item {J} equals two. Hence, frequency patterns involve items {A} and {B}, thus we mine (<A :2, B :2 > |J). As a result, we mine all the frequency patterns of item {J}, which are {B, J :2}, {A, B, J :2}, {A, J:2}.

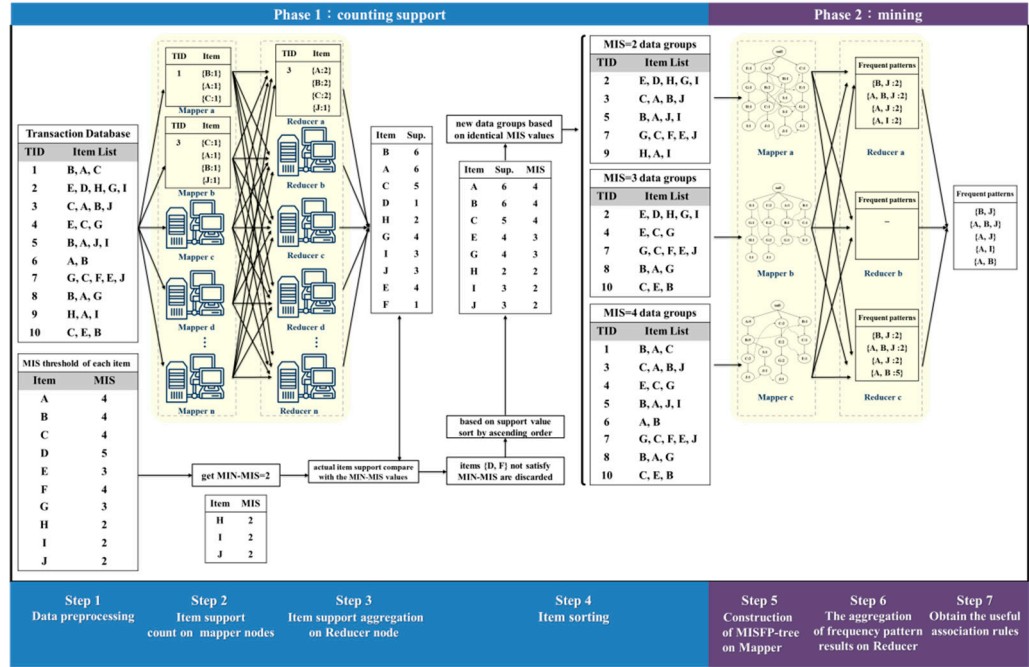

**Figure 3.** Demonstration of the proposed model.

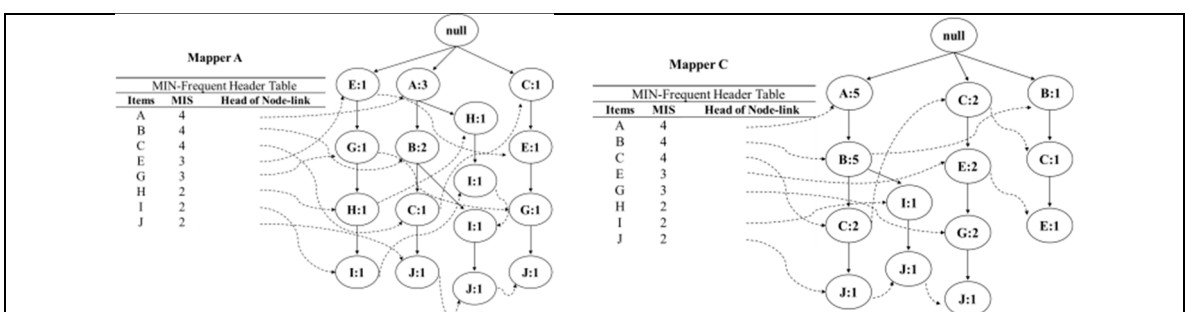

**Figure 4.** MISFP-tree with new data groups in different mapper nodes.

Next, we combine the mined frequent patterns results obtained by all the different mapper nodes. Duplicate items are merged to obtain an aggregate output. According to the data mining output generated by this example, the results of the mining process are shown in Table 5. The most interesting output of the proposed algorithm was its detection of rare patterns. General association rules using a single item support, would also have identify itemsets like {plastic gloves, toothbrush}, {polymeric glass detergent, plastic gloves, toothbrush}, and {polymeric glass detergent, toothbrush}, namely items {A}, {B}, and {J}. However, we also discovered the rare patterns of items {A} and {I}, which represent the polymeric glass detergent and baking soda. This might indicate that baking soda is used to clean glass or other household surfaces.

As demonstrated in the Figure 4, the proposed model was implemented in Hadoop-based environment. In addition, the proposed model enables decision makers set up multiple item support. Therefore, the MISFP-growth can generate valuable patterns of rare item without numerous meaningless patterns.

**Table 5.** The results of frequent patterns.

| Mapper | Suffix Item | MIN-SUP | TID [1] | Conditional Pattern Base | Conditional MISFP-Tree | Frequent Patterns |
|---|---|---|---|---|---|---|
| A | J | 2 | 2, 3, 5, 7, 9 | {A, B, I :1}, {A, B, C :1}, {C, E, G :1} | {A, B :2} | {B, J :2}, {A, B, J :2}, {A, J :2} |
| | I | 2 | | {E, G, H :1}, {A, H :1}, {A, B :1} | {A :2} | {A, I :2} |
| | H | 2 | | {E, G :1}, {A :1} | — | — |
| | G | 3 | | {E :1}, {C, E :1} | — | — |
| | E | 3 | | {C :1} | — | — |
| | C | 4 | | {A, B :1} | — | — |
| | B | 4 | | {A :2} | — | — |
| | A | 4 | | — | — | — |
| B | J | 2 | 2, 4, 7, 8, 10 | {C, E, G :1} | — | — |
| | I | 2 | | {E, G, H :1} | — | — |
| | H | 2 | | {E, G :1} | — | — |
| | G | 3 | | {E :1}, {A, B :1}, {C, E :2} | — | — |
| | E | 3 | | {C :2}, {B, C :1} | — | — |
| | C | 4 | | {B :1} | — | — |
| | B | 4 | | {A :1} | — | — |
| | A | 4 | | — | — | — |
| C | J | 2 | 1, 3, 4, 5, 6, 7, 8, 10 | {A, B, C :1}, {A, B, I :1}, {C, E, G :1} | {A, B :2} | {B, J :2}, {A, B, J :2}, {A, J :2} |
| | I | 2 | | {A, B :1} | — | — |
| | H | 2 | | — | — | — |
| | G | 3 | | {C, E :2} | — | — |
| | E | 3 | | {C :2}, {B, C :1} | — | — |
| | C | 4 | | {A, B :2}, {B :1} | — | — |
| | B | 4 | | {A :5} | {A :5} | {A, B :5} |
| | A | 4 | | — | — | — |

[1] Note: TID is new data group.

## 5. Experiments Results and Analysis

To validate the proposed architecture, three experiments were executed, which tested COI parameters, feasibility, and efficiency accordingly. Appropriate datasets were selected as shown in Table 6. The first experiment used the transaction dataset of groceries [29] with a CFP-growth algorithm to validate the levels of COI. The second experiment used the radioimmunoassay (RIA) dataset, which records the case of hospital. The second experiment verified model feasibility, comparing the consistency of mining results from stand-alone computing. The final experiment used the dataset of retail market basket dataset [30] to evaluate execution time. The results of experiments prove that the

proposed MISFP-growth algorithm can be implemented on the distributed computing framework of Hadoop MapReduce.

**Table 6.** Experimental design, dataset attributes and descriptions.

| No. | Dataset | Volume | Items | Description |
|---|---|---|---|---|
| 1 | Groceries | 9835 | 169 | The Groceries dataset consists of transaction data from grocery store in 2006. Each transaction represents the purchased items. |
| 2 | RIA Report Records | 43,545 | 25 | The RIA dataset consists of hospital case of radioimmunoassay (RIA) in Taiwan from 2009 to 2010. |
| 3 | Retail Market Basket | 88,163 | 16,470 | The dataset covers three non-consecutive periods of supermarket from 1999 to 2000 in Belgian. |

The verification process was based on the virtual operating environment of Oracle VM Virtual Box for parallel architectures, with three versions of the Ubuntu 12.04 LTS operating system, Apache Hadoop 2.2.0 Cluster, and Apache Mahout 0.8. One of the nodes was set as the master node, and two nodes were set as data nodes. Each node had a 3.0 GHz quad-core processor and 8 GB memory capacity. Through the experimental design, we quickly found frequent patterns as well as rare itemsets.

*5.1. Experiment 1: COI Parameters Test*

In Experiment 1 (Exp. 1), we attempted to establish the parameters of COI for the groceries dataset [29]. As shown in Table 7, we set levels of support values by product class and thus reduced the time required to calculate the minimum support for each item. Each itemset belongs to a specific product class, and the support value of any item is equal to that of its product class. For example, the item, "bottled beer", was classified as an "alcoholic beverage (Class 6)" at the tenth level and thus inherited the 4% support value of the category.

**Table 7.** Support value of each product class.

| Levels | Class 1 | Class 2 | Class 3 | Class 4 | Class 5 | Class 6 | Class 7 | Class 8 | Class 9 | Class 10 |
|---|---|---|---|---|---|---|---|---|---|---|
| 10 | 7% | 8% | 9% | 6% | 5% | 4% | 2% | 10% | 3% | 1% |
| | 34 | 35 | 7 | 30 | 21 | 14 | 4 | 15 | 5 | 4 |
| 9 | 7% | 8% | 9% | 6% | 5% | 4% | 10% | 2.5% | 1% | |
| | 34 | 35 | 7 | 30 | 21 | 14 | 15 | 9 | 4 | |
| 8 | 7% | 8% | 9% | 6% | 5% | 4% | 10% | 2% | | |
| | 34 | 35 | 7 | 30 | 21 | 14 | 15 | 13 | | |
| 7 | 7% | 8% | 6.5% | 6% | 5% | 10% | 2% | | | |
| | 34 | 35 | 21 | 30 | 21 | 15 | 13 | | | |
| 6 | 7% | 8% | 9.5% | 6% | 4% | 2% | | | | |
| | 55 | 35 | 22 | 30 | 14 | 13 | | | | |
| 5 | 7% | 2% | 8% | 6% | 5% | | | | | |
| | 58 | 10 | 35 | 30 | 36 | | | | | |
| 4 | 5% | 10% | 7% | 2% | | | | | | |
| | 68 | 22 | 58 | 21 | | | | | | |
| 3 | 5% | 10% | 2% | | | | | | | |
| | 117 | 22 | 30 | | | | | | | |
| 2 | 7% | 3% | | | | | | | | |
| | 122 | 47 | | | | | | | | |

To validate the proposed COI concept in this study, we evaluated the association rules produced in Exp. 1 and found them to be most representative in the third level. Apart from the rare items, frequent patterns at different levels can be found more easily. Furthermore, the average indicator of *lift* was calculated to evaluate the correlations of the association rules. Table 8 counts the association rules mined from the tenth to second levels in Exp. 1 and presents their average *lift*. According to the results of Exp. 1 shown in Table 8, the third level is the optimum parameter for COI because this level

produced more meaningful FP rules from the groceries dataset. Therefore COI = 3 was adopted for Experiments 2 and 3.

**Table 8.** Results of Experiment 1.

| Level | Rules | Lift(avg.) | Level | Rules | Lift(avg.) |
|-------|-------|------------|-------|-------|------------|
| 10 | 139 | 1.670 | 5 | 51 | 1.632 |
| 9 | 22 | 1.574 | 4 | 39 | 1.608 |
| 8 | 23 | 1.490 | 3 | 30 | 1.762 |
| 7 | 23 | 1.619 | 2 | 19 | 1.643 |
| 6 | 19 | 1.597 | | | |

*5.2. Experiment 2: Feasibility Test*

In the Experiment 2 (Exp. 2), using the simple dataset mentioned previously [21] and shown in Table 9a, the MIS thresholds in Table 9b and the RIA dataset were adopted to verify the feasibility of the proposed architecture. The verification process was performed on a virtual machine with a 3.0 GHz quad-core processor and 8 GB memory capacity. One node was set as the master node, and the remaining two nodes were set as data nodes.

**Table 9.** Transaction database, actual support, and MIS of each item for Experiment 2.

| (a) | |
|-----|-----|
| **TID.** | **Transaction** |
| 1 | A, C, D, F |
| 2 | A, C, E, F, G |
| 3 | A, B, C, F, H |
| 4 | B, F, G |
| 5 | B, C |

| (b) | | |
|-----|-----|-----|
| **Items.** | **Support** | **MIS** |
| A | 3 | 4(80%) |
| B | 3 | 4(80%) |
| C | 4 | 4(80%) |
| D | 1 | 3(60%) |
| E | 1 | 3(60%) |
| F | 4 | 2(40%) |
| G | 2 | 2(40%) |
| H | 1 | 2(40%) |

In the first part of Exp. 2, the mined association rules {G, F}, {F, C}, {F, C, A}, {F, B}, {F, A} were consistent for both the CFP-growth algorithm and the MISFP-growth algorithm. In the second part of Exp. 2, 25 items (medical orders) were attributed to the third level of COI, including the hormone test, the hepatitis test, and the tumor marker test. Supports for three COIs were suggested by the National Health Administration and based on the occurrence of cancer in Taiwan. The weightings of rankings were set according to the actual frequency of occurrences in the dataset. Similar items were assigned to the same class, and thus inherited the support value of their COI class, as shown in Figure 5. In the second part of Exp. 2, the medical orders {ABC, ABE, ACHR, ACIGM, B2M, CA125, CA153, FBHCG, FPSA, HAIGM, HAV, HBE, SCC, T3UP, TG, and THY} were abandoned because of they did not satisfy the MIN threshold. The meaningful association rules {TSH, FT4}, {CEA, CA199} were consistent with Wang et al.'s mining association rules [31]. The average lift value was 3.32 and association rules exhibited a positive correlation. These results prove that the proposed model can be applied to the mine association rules with multiple item support thresholds.

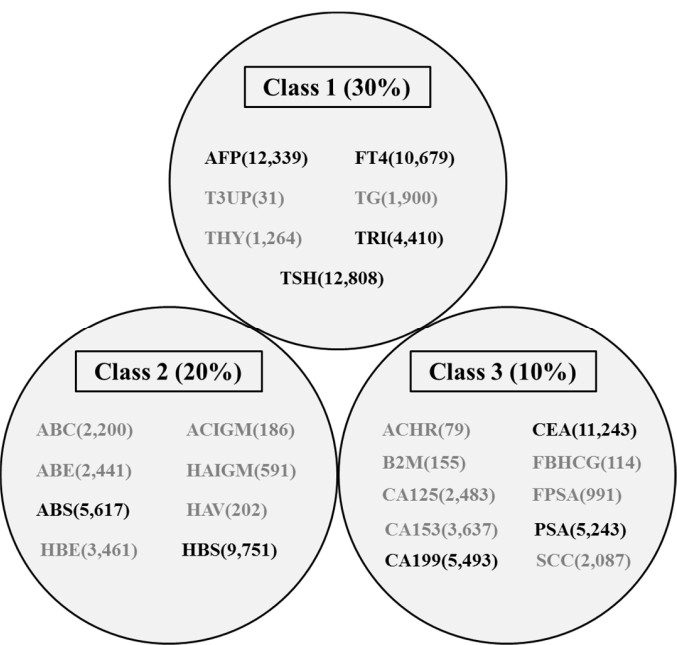

**Figure 5.** Support threshold value of stand-alone operation.

*5.3. Experiment 3: Efficiency Test*

In Experiment 3 (Exp. 3), we tested the efficiency of the proposed algorithm on the retail market basket dataset [30]. The verification process was implemented in five iterations. The experiment environment consisted of a virtual machine with an Ubuntu 12.04 LTS operating system, Apache Hadoop 2.2.0 Cluster, 8 GB memory capacity, and Apache Mahout 0.8. One of the nodes was set as the master node, and two nodes were set as data nodes. Per the experiment results shown in Figure 6, the average execution time of 12,105.6 ms achieved by the parallel architectures represents a reduction of approximately 38% compared to the average execution time of 19,586 ms attained by the stand-alone operation.

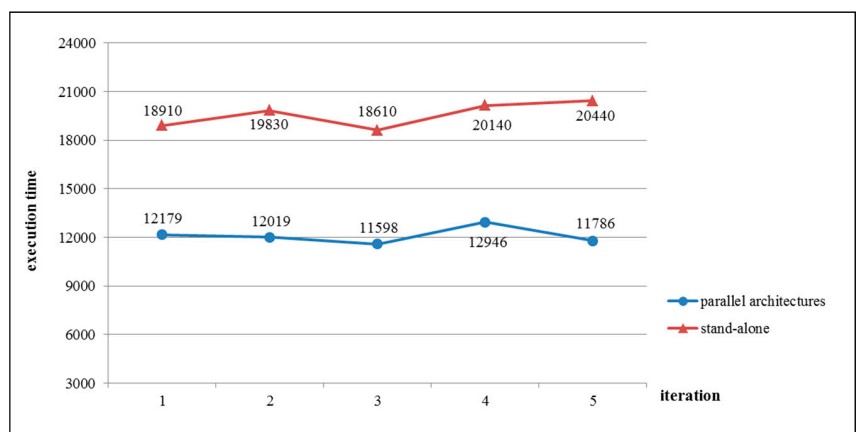

**Figure 6.** Efficiency test of Experiment 3.

These three experiments validated the proposed model, successfully applying the concept of multiple item support thresholds to solve the problems of previous methods that use a single support value. In Exp. 1, the COI concept enabled rapid determination of support thresholds for each item, and the indicator of *lift* showed that the meaningful association rules had a positive correlation on the third level. Experiment 2 verified the feasibility of the proposed model as it identified meaningful association rules that were consistent with the association rules mined in the past research. In Exp. 3,

the average execution time of the parallel architecture demonstrated the improved efficiency of the proposed model compared to the conventional stand-alone architecture.

## 6. Discussion and Conclusions

Big data analytics is changing our daily lives, and increased application of data-driven decision making is inevitable, as the rapid growth in data generation produces new opportunities to extract meaningful information from big data. However, the traditional method of association rule mining from frequent patterns using a single support threshold is not sufficient for today's complex problems and decision making processes. In addition, the efficiency of data analysis must increase to adapt to the rapid growth in data generation.

In this study, a MISFP-growth algorithm consisting of two phases, a counting support phase and a mining phase, is proposed to realize high-efficiency mining of frequency patterns with multiple item supports. To assist decision makers in setting multiple support values for items, we also proposed the concept of the classification of items (COI), which categorizes items of higher homogeneity into the same product class from which the items then inherit their data support threshold. Finally, a correlation indicator, named *lift*, is adopted to evaluate the meaning and accuracy between items and association rules. Furthermore, the MISFP-growth algorithm was implemented without the pruning and reconstruction steps on the distributed computing framework of Hadoop MapReduce.

According to the results from the three experiments, our proposed model can provide decision makers with more relevant frequency patterns. Experiment 2 proved the feasibility of implementing the proposed MISFP-growth algorithm in the Hadoop MapReduce environment. The experiment verified that the same analysis results were obtained from both the stand-alone and parallel architectures. Experiment 3 demonstrated an approximately 38% reduction in the execution time of MISFP-growth algorithm on parallel architectures. Thus, the results of these experiments confirm that the proposed algorithm achieves high-efficiency big data analytics for frequency pattern mining with multiple item supports. In future work, we expect to perform cross-validation by implementing the proposed model in various fields and applications.

**Author Contributions:** C.S. Wang conceived and designed the experiments; J.Y. Change performed the experiments; C.S. and J.Y. analyzed the data and wrote the paper."

**Funding:** This research received no external funding.

**Acknowledgments:** This research was partially supported by the laboratory of enterprise resource planning used for experiments.

**Conflicts of Interest:** The authors declare no conflict of interest.

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
