# Peer review of "MISFP-Growth: Hadoop-Based Frequent Pattern Mining with Multiple Item Support"

_applsci, doi:10.3390/app9102075_

Round 1

Reviewer 1 Report

The introduction can describe more importance and necessary for  consideration of Multiple Item Support while Big Data Analysis applied in practice. Ex. High-profit items mostly with lower selling frequencies.

In section 4, please identify the difference and modification more clear base on Hadoop-based FP Mining Architecture. And then, reader can catch the contribution from this works.

Author Response

Point 1: The introduction can describe more importance and necessary for consideration of Multiple Item Support while Big Data Analysis applied in practice. Ex. High-profit items mostly with lower selling frequencies.

Response: Thanks for your deep suggestions. We have emphasized the importance of the research topic in the introduction section, w. On the other hand, in page 2, we also summarized some previous studies which focused on the same issue to address the trade-offs required by decision makers in the past.

The revisions are listed below.

However, obviously all items in the database vary widely from many perspectives such as their price or profit. Generally, if only one minimum support is used for FP mining, those high-profit items mostly with lower selling frequencies, such as refrigerators and smartphones, would not be considered as frequent patterns [13]. Besides, lower minimum support would lead to the generation of many meaningless association rules, which would complicate the decision-making process. Therefore, using a single minimum support for all items in the database is insufficient for FP analysis. Single item support cannot comprehensively address the complexity of items in large datasets. Thus, it is important and necessary for consideration of multiple item support while big data analysis applied in practice. Recently, several studies have proposed the concept of multiple item supports (MIS) to address the trade-offs required by decision makers [14, 15]. For FP mining research, applying MIS to FP mining warrants further analysis. In addition, lots of researches have attempted to improve the algorithmic efficiency of FP mining when working with big data. Apache’s Hadoop system applies MapReduce programming to solve these problems encountered in processing big data [16]. Using Hadoop MapReduce to implement the parallelization of traditional association rule mining approaches, such as the Apriori and FP-growth algorithms, was proposed to improve the overall performance of frequency pattern mining [17]. For illustration, the k-phase parallel Apriori algorithm was proposed to identify k-frequency items in k-scans using Hadoop MapReduce [18].

Point 2: In section 4, please identify the difference and modification more clear base on Hadoop-based FP Mining Architecture. And then, reader can catch the contribution from this works.

Response: Thanks for your useful suggestions. In section 4, we have indicated the difference in page 6 and listed below.

--------------------------------------------------------------------------------------------------------------------------

As demonstrated in the Figure 4, the proposed model is implemented in Hadoop-based environment. In addition, the proposed model enables decision makers set up multiple item support. Therefore, the MISFP-Growth can generate valuable patterns of rare item without numerous meaningless patterns.

Reviewer 2 Report

Thi paper deals with Hadoop-based Frequent Pattern Mining with multiple item support.

The following items are considered:

(1) The more good title of the paper is suggested as "MISFP-growth: Hadoop-based Frequent Pattern Mining with Multiple Item Support"

(2) Figure 1 have to be deleted and have to be replaced by simple sentences which describes each method. So, it is good to describe problems in these methods.

(3) Authors have to give comparison results between the proposed method and previous methods with respect to efficiency and correctness.

(4) Specially, authors have to suggest good points of the paper by giving comparison results.

(4) English have to be checked.

Author Response

Point 1: The more good title of the paper is suggested as "MISFP-growth: Hadoop-based Frequent Pattern Mining with Multiple Item Support"

Response: Thanks for your suggestions. The paper title has been revised.

Point 2: Figure 1 have to be deleted and have to be replaced by simple sentences which describes each method. So, it is good to describe problems in these methods.

Response: Thanks for your suggestions. Figure 1 has been removed. Instead, we have described the main idea each algorithm in Table 1. Please see page 3 for your reference.

Point 3: Authors have to give comparison results between the proposed method and previous methods with respect to efficiency and correctness. Specially, authors have to suggest good points of the paper by giving comparison results.

Response: Thanks for your suggestions. We have further detail the purpose of each experiment in page 8. Specifically, the second experiment is a correctness test and the third experiment shown the efficiency of the proposed model. The experiment design and analytics results of three experiments are detail in Section 5. Please find page 8 to page 11 for your reference.

Point 4: English have to be checked.

Response: Thanks for your reminder. The manuscript has been modified by English native speakers.

Round 2

Reviewer 2 Report

I think that authors revised the paper according to my comments.

So, the paper is enough to be published. Before the paper is released, English grammar have be checked again even of it was reviewd by native speaker.